# Enhancing Urban Mobility through Traffic Management with UAVs and VLC Technologies

Javier Garau Guzman [1] and Victor Monzon Baeza [2,*]

1. Department of Computer Science, Multimedia and Telecommunications, Universitat Oberta de Catalunya, 08018 Barcelona, Spain; jgaraug@uoc.edu
2. SigCom Group in SnT, University of Luxembourg, L-1855 Luxembourg, Luxembourg
* Correspondence: victor.monzon@uni.lu

**Abstract:** This paper introduces a groundbreaking approach to transform urban mobility by integrating Unmanned Aerial Vehicles (UAVs) and Visible Light Communication (VLC) technologies into traffic management systems within smart cities. With the continued growth of urban populations, the escalating traffic density in large cities poses significant challenges to the daily mobility of citizens, rendering traditional ground-based traffic management methods increasingly inadequate. In this context, UAVs provide a distinctive perspective for real-time traffic monitoring and congestion detection using the YOLO algorithm. Through image capture and processing, UAVs can rapidly identify congested areas and transmit this information to ground-based traffic lights, facilitating dynamic traffic control adjustments. Moreover, VLC establishes a communication link between UAVs and traffic lights that complements existing RF-based solutions, underscoring visible light's potential as a reliable and energy-efficient communication medium. In addition to integrating UAVs and VLC, we propose a new communication protocol and messaging system for this framework, enhancing its adaptability to varying traffic flows. This research represents a significant stride toward developing more efficient, sustainable, and resilient urban transportation systems.

**Keywords:** traffic management; VLC; UAV; automation; smart city





## 1. Introduction

In the modern era of urbanization, the concept of "smart cities" has emerged as a beacon of hope for addressing the myriad challenges posed by growing metropolitan populations. A smart city represents an integrated urban ecosystem that harnesses the power of advanced technologies to enhance the quality of life for its inhabitants. At the heart of this transformation lies an innovative approach to mobility management, a critical facet of urban life. Smart cities strive to revolutionize how people move within and across urban spaces with the goal of creating more efficient, sustainable, and accessible transportation networks, meeting the 2030 sustainable development goals [1].

The traditional paradigms of urban mobility are being redefined through the infusion of cutting-edge technologies. From intelligent transportation systems and real-time data analytics to the integration of autonomous vehicles and sustainable transportation modes, smart cities are revolutionizing how individuals navigate their urban environments. By seamlessly weaving together modes of transport, optimizing traffic flows, and prioritizing accessibility for all, smart cities aim to alleviate congestion, reduce emissions, and enhance overall livability. Urban centers across the globe are grappling with escalating challenges in traffic congestion and mobility. As populations continue to surge, traditional traffic management systems encounter limitations in regulating flow and inefficiently alleviating congestion. This escalating urban mobility crisis necessitates innovative approaches integrating advanced technologies to revolutionize traffic management. In response to this imperative, this paper delves into the synergistic application of Unmanned Aerial

Vehicles (UAVs) and Visible Light Communication (VLC) technologies to augment urban traffic management.

The evolution of urban mobility solutions has witnessed remarkable strides in recent years. Unmanned Aerial Vehicle (UAV) advancements have expanded their potential beyond recreational and surveillance applications. UAVs are now pivotal instruments in real-time data acquisition and monitoring, offering unprecedented access to crucial information from elevated vantage points. Furthermore, the emergence of Visible Light Communication (VLC) technology has introduced a transformative communication framework. This technology leverages the visible light spectrum, relieving congestion produced in the radio frequency spectrum and opening new avenues for reliable, high-speed data transmission. While both UAVs and VLC technologies individually hold promise, their combined application in urban traffic management remains an emerging frontier with immense potential.

### 1.1. Background

In the rapidly evolving landscape of urban mobility, the integration of cutting-edge technologies has become imperative for addressing the complex challenges posed by growing urbanization [2–4]. A diverse array of studies, each offering unique perspectives and insights, contribute significantly to the discourse on enhancing urban mobility [5]. This compilation of research endeavors encompasses a broad spectrum of topics, ranging from traffic management strategies to the potential applications of Unmanned Aerial Vehicles (UAVs) [6] and Visible Light Communication (VLC) [7] technologies. While certain studies focus on the optimization of traffic control systems using connected and automated vehicles (CAVs) [8,9], others venture into the realm of urban air mobility, proposing hierarchical planning procedures for managing fleets of aerial vehicles [10]. Additionally, the research delves into the potential of VLC-enabled UAVs, contemplating their power-efficient deployment for communication and illumination purposes [11]. However, it is important to note that not all studies explicitly explore the utilization of UAVs and VLC technologies for urban mobility enhancement. Additionally, not only traffic lights, as considered in this work, but VLC-enabled streetlights [12] can also play a pivotal role in enhancing urban mobility and traffic management, ensuring seamless connectivity and communication between vehicles and infrastructure.

Furthermore, a broader perspective emerges from studies addressing the fundamental challenges facing smart cities. Safety, privacy, ethical, and legal concerns surrounding the integration of UAVs in smart cities are aptly examined [13], underscoring the multifaceted considerations accompanying technological advancements. Additionally, the pivotal role of intelligent transportation systems (ITS) and artificial intelligence (AI) in optimizing urban planning and predicting traffic conditions is explored, shedding light on the potential for data-driven solutions in shaping the future of urban mobility [14]. Moreover, the papers collectively highlight the increasing significance of urban air traffic management systems in accommodating the rising number of UAVs in smart cities [15]. The focus is on harnessing the potential of CAVs to provide valuable data for traffic management and actively improve traffic flow. For example, simulations conducted in [9] for a real-world arterial corridor show significant mobility and fuel economy improvements, with total delay reduced by 2.2% to 33.0% and fuel consumption lowered by 3.9% to 7.4%. A groundbreaking approach to traffic control in smart cities is presented in [16], utilizing multiple UAVs for enhanced event detection.

In traffic congestion detection, the choice of the right algorithm plays a pivotal role in ensuring accuracy and efficiency. In this context, we have opted for the You Only Look Once (YOLO) algorithm as our preferred method for congestion detection, standing out among other available algorithms such as SSD and Faster R-CNN. YOLO presents a unique and promising approach to real-time detection for traffic detection application [17]. Its ability to analyze entire images in a single pass provides a swift and comprehensive understanding of the traffic scenario. This introduction delves into the rationale behind selecting YOLO

over alternative algorithms and sets the stage for a detailed exploration of its capabilities in revolutionizing congestion detection within urban environments. Here are the reasons for choosing YOLO:

1. **Real-Time Processing Speed**: YOLO processes frames at high speed, ranging from 45 frames per second (fps) for larger networks to 150 fps for smaller networks. This real-time processing capability is crucial for applications like traffic management, where timely detection and response are essential. YOLO's speed advantage ensures swift processing of video frames, enabling rapid decision-making in dynamic traffic scenarios.

2. **Efficiency in Resource Usage**: YOLO is known for its efficiency in resource utilization, making it well-suited for deployment on resource-constrained devices like UAVs. While it may have comparatively lower recall and increased localization error, the trade-off is acceptable in scenarios where real-time processing and efficiency are prioritized.

3. **Single-Pass Object Detection**: YOLO follows a single-pass object detection approach, dividing the input image into a grid and directly predicting bounding boxes and class probabilities. This design contrasts with two-pass methods like Faster R-CNN. The single-pass architecture aligns to minimize processing time and resource usage.

4. **Simplicity and Ease of Integration**: YOLO's simplicity and straightforward architecture make implementing and integrating into the overall system easier. The streamlined design contributes to faster inference and facilitates deployment on UAVs with limited computational capabilities.

5. **Suitability for UAV Applications**: Considering the use case involving Unmanned Aerial Vehicles (UAVs) for traffic management, YOLO's balance between speed and accuracy aligns with the requirements of real-time processing images captured by UAVs.

Table 1 illustrates YOLO's superior precision and a higher count of true positives compared to alternative models. Although its frames-per-second processing speed falls between that of SSD and other models, we prioritize detection precision for our system. Since SSD sacrifices precision in our specific context of identifying cars, we emphasize detection accuracy in our evaluation. These data are from an example of car detection on the road in [17].

**Table 1.** YOLO comparison with other algorithms for detection.

| Model | Average Precision | True Positive | False Positive | Frame per Second (FPR) |
|---|---|---|---|---|
| YOLO v4 | 98.08% | 273 | 25 | 82.1 |
| R_CNN | 93.2% | 262 | 17 | 36.32 |
| SSD | 92.7% | 257 | 34 | 105.14 |

In the context of [18], these identical models underwent a comparison for an alternative image detection scenario, specifically involving version 3 of YOLO, where the latter exhibited a superior performance compared to other models. This scenario holds significance as it involves substantial losses in the link due to the considerable distance between the satellite and the ground. Such findings are relevant as they can be extrapolated to situations with low visibility caused by factors such as fog, pollution, or other environmental conditions. On the other hand, YOLO has presented a better performance in terms of processing speed than RetinaNet, according to [19]. In the present era, chip technologies, such as AI accelerators based on GPUs, provide extensive signal processing advantages in diverse scenarios. Notably, these scenarios encompass nanosatellites, CubeSats, and, of particular relevance to our discussion, Unmanned Aerial Vehicles (UAVs), as elucidated in the recent reference [20]. The processing demands mandated by the YOLO algorithm align adequately with the application at hand.

As the demands on urban transportation continue to escalate, these studies collectively underscore the urgency to leverage emerging technologies and implement forward-thinking strategies to create sustainable, efficient, and resilient urban mobility solutions. Through a multidisciplinary approach that embraces technological innovation, policy considerations, and integration with broader urban planning frameworks, the quest for enhanced urban mobility takes on a collaborative and holistic perspective. This compilation serves as a testament to the dynamism of research in the field. It provides a valuable foundation for future endeavors to transform urban mobility into a safer, more efficient, and sustainable reality.

*1.2. Our Contribution*

In light of this context, given the backdrop of smart cities, urban mobility, and the imperative for enhanced traffic management, our contribution encompasses the following key facets:

1.  Identifying traffic patterns from analyzed images and utilizing a Congestion Traffic Algorithm (CTA) to ascertain the presence of congestion.
2.  Formulating a message format protocol rooted in variable length, facilitating seamless information exchange among the components of the proposed system.
3.  Creating a communication protocol that enables system elements to engage with one another and assists them in determining their operational states. This is crucial in ensuring that the system functions optimally within the dynamic context of smart cities and contributes to the broader objective of improving urban mobility through efficient traffic management.

This work continues the contribution initially proposed as an academic work in [21].

The rest of this paper is organized as follows. The proposed traffic management prototype system is presented in Section 2. The algorithms integrated into the prototype are defined in Section 3. The message format protocol and the communication protocol are exposed in Section 4. Prototype insight evaluation is displayed in Section 5. Finally, the conclusions are presented in Section 6.

## 2. Traffic Management Prototype System

In this section, we explain the system architecture proposed to manage traffic through VLC and UAV technologies, as well as the interconnection of all the elements in our system.

*2.1. Overview System Architecture*

Our system will be composed of four elements, each with a different function and task. These four elements are the base station, supervisor UAV, UAV detector, and traffic light corresponding to the numbers 1, 2, 3, and 4, respectively, in Figure 1. These elements are linked by bidirectional links, except in the case of those generated between the UAV detector and the traffic light, which will be unidirectional.

Next, we will present each of the elements of our proposed system shown in Figure 1 and the function they realize.

1.  *Base station*: Our first system element is the base station. This station must be located on high ground or close to the supervised traffic area since direct vision is required in VLC technology. Likewise, UAVs must return to a safe place where they can recharge their batteries or be attended to in case of failure. This is why the location becomes important because UAVs should not travel a long distance. The access to this location should also be controlled to avoid intruders or incidents. The base station's main function is to send and receive information from the supervisor UAV, so we have a bidirectional link between these two elements.

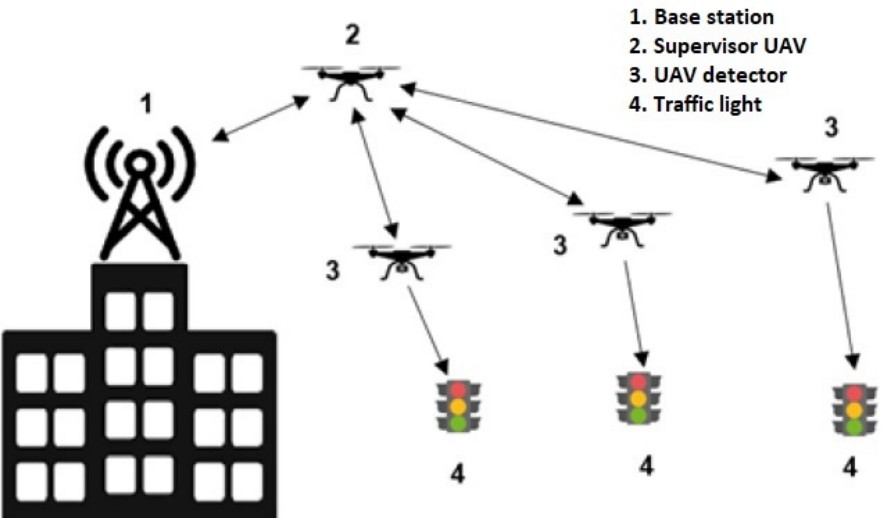

**Figure 1.** Proposed system architecture.

2. *Supervisor UAV*: This supervisor UAV will link the UAVs monitoring the traffic and traffic lights between the base station. Due to this, we will have bidirectional links to the base station and UAV detector.

   This element provides us with two main functions. The first one is to inform the base station of the status of the detector UAVs and traffic lights. The second function will allow us to modify the traffic congestion parameter associated with the detector UAVs to make our system more sensitive to traffic jams.

   In case a supervisor UAV must return to the base station at any time due to an incident or low battery, the detector UAVs could operate autonomously. The only temporarily unavailable features would be the information updates regarding traffic lights and detector UAV status, as well as the ability to adjust the traffic congestion parameter for the detector UAVs. These two specific functions will be reinstated upon their return to the base station or when a supervisor UAV re-establishes connection. This decision ensures that the service remains unaffected in case of an incident, and supervision can be restored by deploying a new supervisor UAV.

3. *Detector UAV*: This element is a key piece of the architecture as it is tasked with executing the three fundamental functions for traffic regulation, which are as follows:

   (a) Taking aerial photographs.
   (b) Detect traffic intensity.
   (c) Notify the traffic light of the need to change its operating mode.

   The initial function involves capturing aerial images of streets or intersections controlled by the traffic light. For this purpose, the UAV will be equipped with a camera responsible for this task. The image resolution must meet the minimum quality standards. In other words, if the image is not sufficiently clear for vehicle identification, accurate detection and proper traffic management would be compromised. If the UAV has to supervise more than one traffic light, it will have the supervised traffic light map as configuration, and each one of the traffic lights will have its address to receive information from the UAV.

   The second function of the detector UAV is to detect the intensity of the apparent traffic in it once the photograph is taken to know if there is saturation. To do this, the UAV will have an object detection algorithm. After the image processing by the UAV, it uses parameters extracted from the image to compare the occupancy of vehicles on the road with certain predefined values. Depending on this comparison, the UAV will notify the traffic light if it is necessary to change the configuration of the red and green lights and the time they must remain active. Within the value of the

congestion parameter extracted from the image, it will be located within certain ranges of values that will be defined for the system so that if the parameter is contained in a certain range, both the UAV and the traffic light must be found in a given mode of operation. In the same way, if there is a change in the image parameter and it enters a new range, the UAV and the traffic light may have to make a mode change to adapt to the latest traffic conditions.

The third function is to notify the traffic light of the need to change the operating mode to adapt to a new traffic situation. To do this, through VLC technology, the detector UAV will inform, using our defined protocol, the traffic light that must make the change.

Because of the functions exposed to this, we will have bidirectional links to the UAV supervisor and a one-way link to the traffic light.

In Figure 2, we can observe the operational process of a detector UAV as depicted in its flow chart. The sequence initiates with the UAV capturing an image. Subsequently, the UAV analyzes to determine whether there is congestion or vehicle saturation within the monitored area. If not, the UAV will reattempt image capture after a designated waiting period and proceed with a fresh analysis. However, if saturation is detected, the detector UAV will transition to a different mode and relay the necessity for a mode shift to the associated traffic light. Upon notifying the traffic light, another waiting period ensues. The specific mode of operation determines this duration before the UAV proceeds to capture another image.

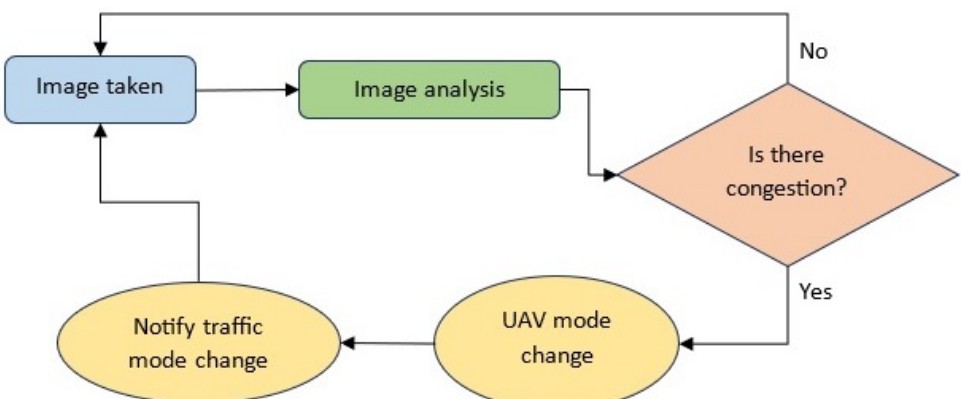

**Figure 2.** Flowchart of detector UAV.

4.  *Traffic light*: This component controls the intersection and manages vehicle flow through its signals. In our configuration, it operates as a passive device, receiving information and creating a one-way link from the detector UAV.

    The traffic lights will operate in two primary modes: autonomous and remote. In autonomous mode, a traffic light controller will regulate the traffic lights, as is traditionally done. Conversely, in the remote mode, changes in the traffic lights will be guided by a detector UAV based on captured and processed information.

*2.2. Technologies*

We have the links that interconnect the elements of our proposed system as shown in Figure 1. These links allow communication between all the elements, and they are based on VLC technology. Bidirectional VLC links will enable us to communicate with the supervisor UAV with the base station and UAV detector. Downward VLC links connect the UAV detector and the traffic lights because traffic lights are a passive element in our system prototype and they react according to the decision of the UAV detector.

Looking at Figure 3, we show the diagram that allows us to visualize the elements of the system along with the types of networks involved in the system and the technologies used. As can be seen, there will be two networks in the system, one formed by the data

backbone network and the other being the UAV network formed by UAVs, which we can see in the yellow and green circles. The technology that will be used in the backbone network will be IP, while in the UAV network part, it will be through VLC, which is depicted in Figure 3, as a red line for the IP part and a blue line for the VLC connections. As can be seen, the UAV network will be composed of all the detector and supervisor UAVs that are part of the system and the UAV will be connected to traffic lights and the base station associated via VLC. For its part, the base station serves as a junction point between the data backbone network and the UAV network since it will change the information from VLC to IP and vice versa.

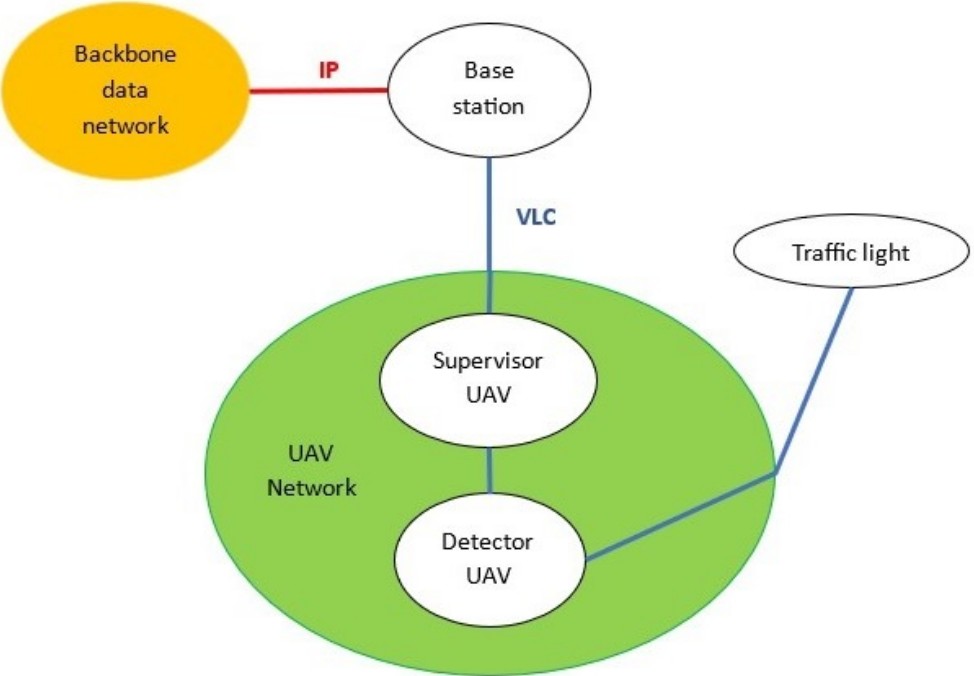

**Figure 3.** Technology used and system network types.

Regarding how the elements of the system will have access to the environment, this will be carried out through Time Division Multiple Access (TDMA). TDMA preserves all the bandwidth of the channel, dividing it into alternating time spaces with which each element may transmit in its assigned space. In turn, multiplexing in time has a lower battery cost than frequency multiplexing, and due to this, we consider TDMA as a better choice to be used in UAVs. Likewise, multiplexing in time is easier than in frequency.

*2.3. Element Interconnection*

- Base station: we can take a look at Figure 4 to see how the base station is interconnected.

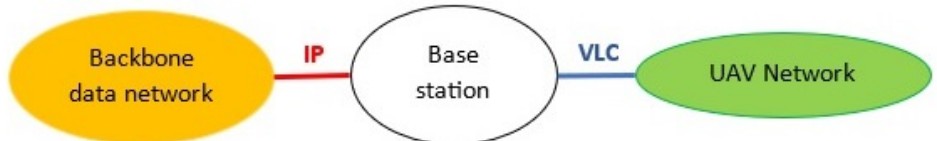

**Figure 4.** Connections of the base station.

In order to establish VLC communication between the base station and the supervisor UAV, our base station has to have a VLC transmitter and receiver. On the part of the backbone network, we will not have problems since, in this case, with any network access device, we could send and receive data if we have access to them.

Regarding the issue of powering the systems, as we are working with a fixed installation with access to an electrical supply, we should not have any problem powering our

devices, which are the VLC transmitter, VLC receiver, and battery charging stations for UAVs.

- Supervisor UAV: as we see in Figure 5, the supervisor UAV has to have a VLC transmitter and receiver on board to be able to communicate with the base station and the detector UAV. Due to the supervisory function, it is not necessary for it to have any other gadget on board; thus, we can reduce its weight and thereby increase flight time.

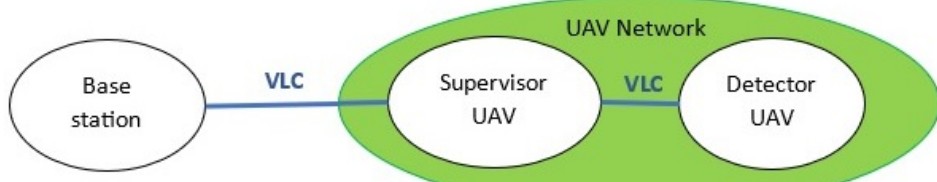

**Figure 5.** UAV supervisor connections.

- Detector UAV: as we exposed in Section 2.1, the UAV detector has to notify the traffic light of the need to change the operating mode to adapt to a new traffic situation. To do this, through VLC technology, the detector UAV will inform, using our defined protocol, the traffic light that must make the change. Attending to hardware issues, this UAV will have on board a high-definition camera that allows us to obtain the images to be processed, as well as a VLC transmitter and receiver. In Figure 6, we can observe the connections of this UAV through VLC with the other system elements with which it will communicate.

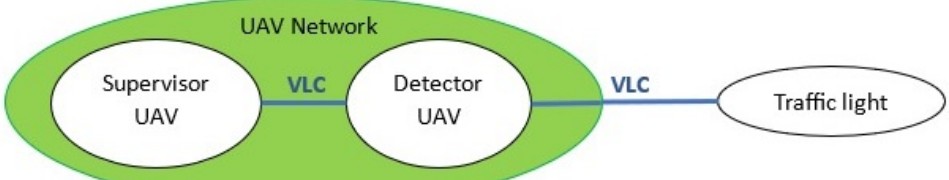

**Figure 6.** UAV detector connections.

- Traffic light: this component functions as a passive device, solely receiving information. Consequently, we will require a VLC receiver linked to the detector UAV responsible for its remote operation.
  The power supply for the traffic light is easily managed, as we have access to the electrical grid. The transition between autonomous and remote modes can be executed through a relay triggered by the traffic light. This arrangement enables the control signal from the traffic light regulator to be interrupted when the traffic light needs to operate remotely.

## 3. Algorithms Integrated in the Prototype

In this section, we present two key aspects. Firstly, we detail the process by which images captured by the UAVs will be analyzed. Secondly, we outline the method for identifying traffic saturation within the monitored area.

### 3.1. Image Processing Algorithm

There are multiple approaches to object recognition, with some machine learning (ML) and deep learning (DL) techniques having become generalized approaches for object recognition problems. ML and DL allow us to learn to identify and recognize objects in images, but their execution makes them different. In [22,23], we can see how both approaches or techniques detect objects.

According to [22,23] in order to perform object recognition using ML, images or videos must be collected and the relevant features of each of them must be selected so that, for example, a feature extraction algorithm can obtain information from these images and videos from edges or corners to differentiate different kinds of data. Based on these characteristics, an ML model would be applied to classify the data into different categories in order to use the information obtained in the analysis and classification of new objects.

For DL, and looking back to [22,23], convolutional neural networks (CNN) are used to automatically learn the characteristics of a certain object before detecting it. Within DL we can find two approaches when it comes to object recognition. These two approaches consist of training a model from scratch or using a pre-trained DL model. In the first approach, which consists of training a model from scratch, a large amount of labeled data must be collected, and the network architecture must be designed to learn and create the model. This method requires more time. In the second, we start with a previously trained model, applying a transfer learning approach. This existing model is provided with new information that will contain previously unknown classes. This second method does not need as much time as the previous one to deliver the results since the model is previously trained with thousands or millions of images.

For object detection in images, an algorithm should be applied on board the UAV that allows this action. For our developed system, we choose a DL approach because it is more autonomous than ML, in the same way that it is less complex and sophisticated, and once the system has been programmed, it will practically not require human intervention. We also decided to use an already trained model since UAVs may not have enough autonomy to be able to efficiently offer traffic management service if they have to pass through a training phase.

Now that it is known that we will use a previously trained model of DL, we have to define the algorithm that will be used for object detection in the images. Our system can use the YOLO algorithm. According to [24], YOLO consists of an open-source system of the state of the art for real-time object detection using a single CNN that only needs to view the image once.

### 3.2. Traffic Congestion or Saturation Detection

We will look for a way to define whether the monitored road is saturated or congested based on the image processing carried out through a CTA.

As we have seen in [24], YOLO allows us to detect different kinds of objects in the same image, but we need to be able to know if these objects are causing the traffic or if, on the contrary, they are objects that have nothing to do with traffic. To know this, we must go down to the level of the object detection and classification functions of the YOLO algorithm. With this, we will try to use the coordinates of the bounding boxes of each of the objects detected in the image by YOLO, and only the areas of those that are of interest to our system will be calculated. For all those vehicles, calculate the total area of vehicles that make up the image and compare it with the total area of the same to detect the existing traffic load. Now that it has been explained how the saturation of a certain pathway will be detected, the procedure by which we can obtain all this information will be explained.

Looking at Figure 7, we observe bounding box predictions using the following parameters: $p_c$, $b_x$, $b_y$, $b_h$, $b_w$, and $c$. The $c$ parameter indicates the class of object detected in the bounding box. The next ones are the parameters $b_x$ and $b_y$, which are equivalent to the mathematical coordinates referring to the center of the box regarding the location of the cell in the bounding box. Finally, we have the parameters $b_h$ and $b_w$, which are equivalent to the height and width of the dimensions of the bounding box. The $p_c$ parameter consists of the security that an object is present in the bounding box, thus, we have the probability that we saw in the previous section that this object is being detected and classified in the correct category.

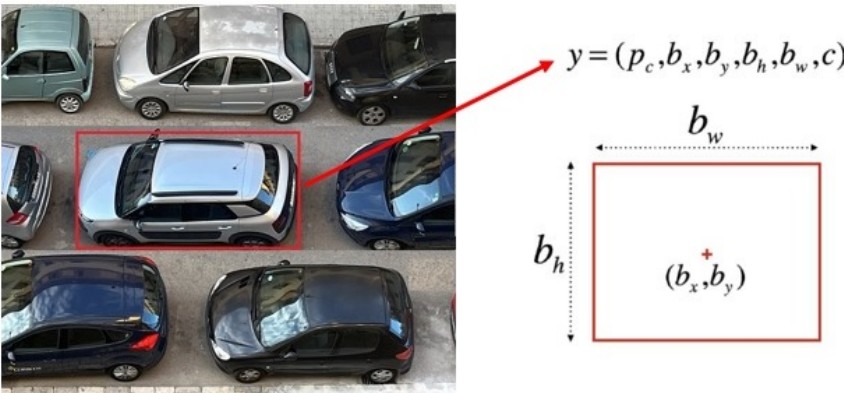

**Figure 7.** YOLO parameters for object detection.

To calculate that detection confidence, the algorithm must contain an object that is of some class in the bounding box. Thus, the calculation can be seen in Figure 8, where, based on Figure 7, it is observed that a car is detected. In the upper array of Figure 8, the parameters discussed in the previous paragraph can be seen, followed in the same array, in red, by the possible types of classifiable objects. It can be seen in this same figure that to detect the type of object, $p_c$ is multiplied by the array of possible defined object types. The type of object selected is the one that has greater security, in the case of the example object three, which corresponds to a car. In this way, we have learned how to select the type of detected object, and from here, it will be explained how the occupancy of the detected objects that influence traffic will be calculated.

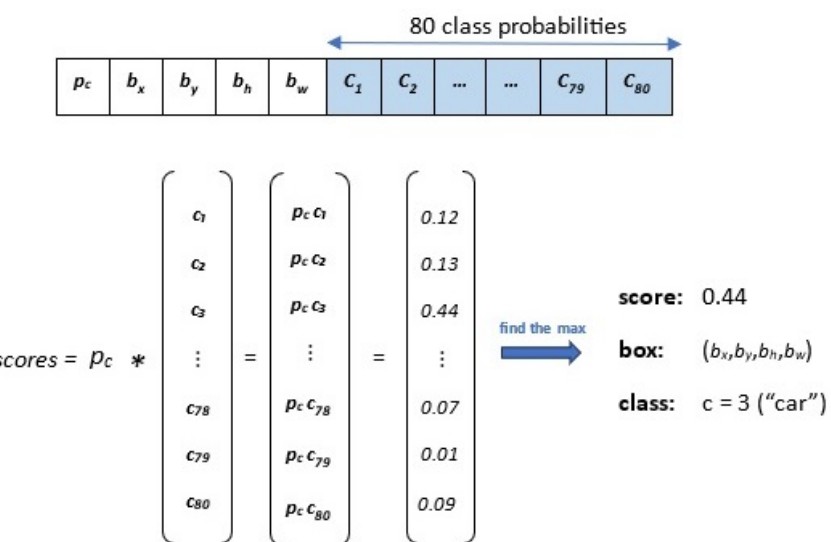

the box $(b_x, b_y, b_h, b_w)$ has detected c = 3 ("car") with probability score: 0.44

**Figure 8.** Object type probability calculation by YOLO.

As we mentioned in Section 3.1, the input image will be resized to $448 \times 448$ pixels. Starting from here and knowing the parameters that define each bounding box, the area occupied, which in this case, is the pixels occupied by the detected and classified objects that affect traffic, can be calculated. This means we will only consider objects classified as vehicles, such as cars, motorcycles, buses, or trucks. Knowing the value of the parameter $c$ that characterizes each image, the image can be filtered to know how many objects with these characteristics are on the road. In the same way, we can discard those identified objects that we are not sure whether their classification has been correct, since if, for example, a car

is detected with less than ten percent security, this detection may not be correct, or it is an object type that is not interesting for our system.

We have already seen that we can obtain and filter data by knowing the parameter of c and the types of detected objects that interest us. To calculate how much image space each occupies, we must first know how many pixels each bounding box occupies.

As seen in Figure 9, an illustrative object is detected in the 448 × 448 pixel image. In this figure, in the same way, we can see $x$, $y$, $w$, and $h$, which will correspond to $b_x$, $b_y$, $b_h$, and $b_w$, which we have mentioned previously.

The detected object is centered at pixel coordinates 185 on the x-axis and 189 on the y-axis. Additionally, the bounding box exhibits a width of 220 pixels and a height of 99 pixels. We found detailed information corresponding to these measurements on the right side of the image, as described earlier. Specifically, we have the dimensions of the bounding box, enabling us to calculate its area. For rectangles, the area is determined by multiplying the width by the height. In the case of Figure 9, the area covered by the bounding box is calculated as 220 × 99 pixels, resulting in an area of 21,780 pixels. To ascertain the proportion of the entire image occupied by the bounding box, we consider that the image encompasses 448 × 448 pixels, totaling 200,704 pixels. We obtain the occupancy percentage by dividing the bounding box's area by the image's total area. For Figure 9, the bounding box representing the grey car inside the red box, covering approximately 0.092 of the total image area, and assuming the total is 1, this signifies an occupancy of 9.2%.

**Figure 9.** Parameters detected by YOLO in an image.

To extrapolate this method to our system, we must first filter and consider, using parameter $c$, those objects of interest that have already been mentioned as cars, motorcycles, buses, or trucks. Next, the area of all the objects in these categories must be calculated, and the space that these objects are occupying on the total image must be detected. In this way, a value will be obtained, which in the case of Figure 9 is 0.092, with which it will be possible to know the occupancy of vehicles on the road.

We rely on the occupancy value of the image analysis to gauge the extent of the congestion. This value serves as the basis for establishing a meter segmented into various levels which governs the operational states of our system's components. Illustrated in Figure 10, this segmented meter is delineated by distinct lower and upper thresholds, indicating the degree of vehicular occupancy on the road. As depicted, the meter progresses from left to right, demarcating varying segments corresponding to different occupancy levels.

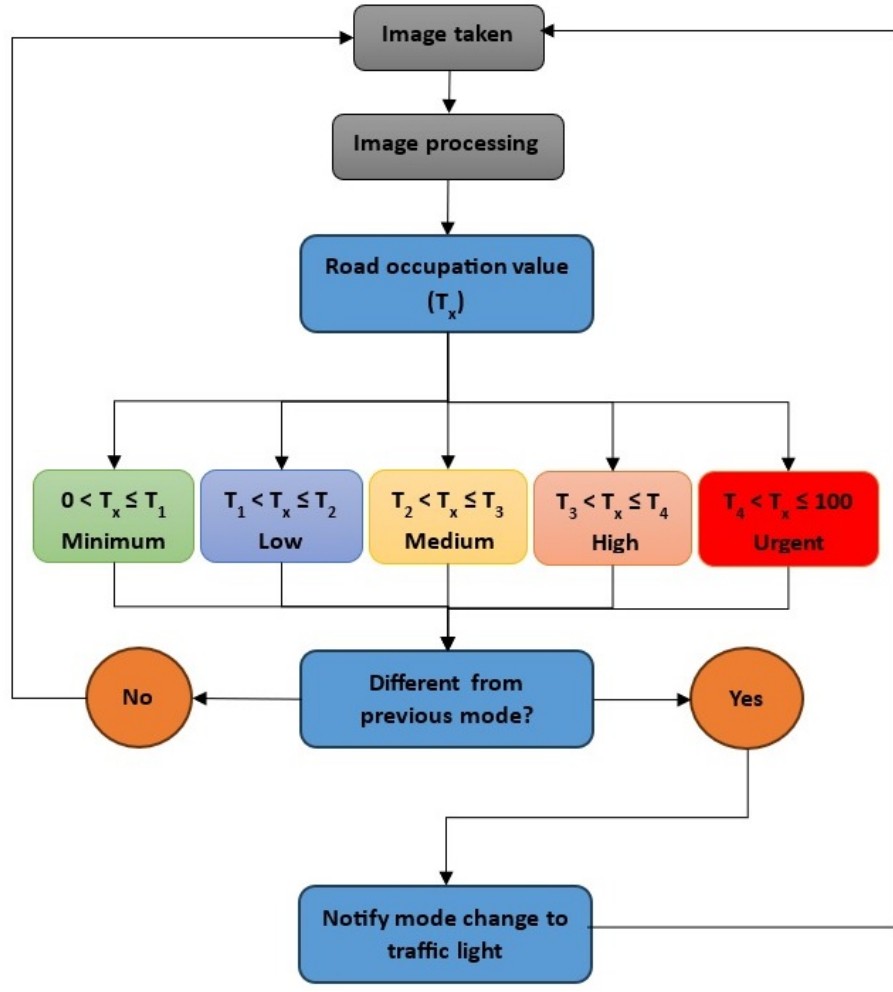

**Figure 10.** Thresholds and levels to indicate the occupancy on the road.

To visually understand the operation of the proposed CTA, see Figure 11. In this figure, we can see the steps taken to know if there is a saturation of vehicles or not. The way this is done can be divided into three significant steps:

- **Step 1:** UAV takes an aerial photograph and processes it. After this, we will obtain a road occupation value $T_x$.
- **Step 2:** $T_x$ value obtained is compared to the limit threshold values of the mode segments to know at which mode $T_x$ is located.
- **Step 3:** once $T_x$ defines the mode we are in, it is compared to the previous mode detected by the UAV. In case they are not the same, the UAV will notify the traffic light it has to change to the mode where $T_x$ is located and, after that, return to take images. On the other hand, if the previous mode is the same, the UAV returns to take a photograph.

**Figure 11.** Flowchart of CTA.

The threshold values of each mode can be adjusted or modified by an operator or administrator. This way, we can make the system more sensitive to traffic since we will modify the thresholds for each mode, and the occupation of the road could be detected in different ways. The proposition of two possible scenarios with the result obtained in

Figure 9, where we have seen that the occupancy value obtained was $T_x = 0.092$, can be taken to show how the system changes according to the threshold values.

**Scenario 1: System less sensitive to traffic:**

In the first scenario, we assume that an operator has set the threshold values to $T_1 = 0.2$, $T_2 = 0.4$, $T_3 = 0.6$, and $T_4 = 0.8$. With these values inserted, CTA will compare $T_x = 0.092$ with the limit threshold values of the segments and detect the minimum mode due to $T_x = 0.092$ being between 0 and $T_1$.

**Scenario 2: System more sensitive to traffic:**

However, in the second scenario, we propose that another operator set the threshold values as $T_1 = 0.05$, $T_2 = 0.15$, $T_3 = 0.3$, and $T_4 = 0.6$. Once these new values are inserted, CTA compares the value of $T_x = 0.092$, detecting a low mode because $T_x = 0.092$ is settled between $T_1$ and $T_2$.

This leads us to an effective CTA that provides insights into the level of vehicular occupancy on the road and facilitates the calculation of their overall occupancy. This is achieved by establishing distinct segments corresponding to different levels of occupancy. The thresholds we have introduced play a pivotal role in delineating operational modes for our system's components, and an operator or administrator can adjust them for optimal performance.

## 4. Protocols

In this section, a protocol will be defined that allows the devices that make up the system in Figure 1 to communicate, introducing and explaining how we can find all the elements of our system.

### 4.1. Message Format Protocol

Firstly, a protocol is defined as the rules and conventions applied between two peer entities to carry out a certain function or service. In our case, these entities will be the base station, the supervisor UAV, the detector UAV, and the traffic light, and we will define the rules through which they will communicate with each other. Within the protocols, there are two types: connection-oriented and connection-free. The main difference between the two is based on the fact that in the first case, a connection is established between the sender and the receiver before data transmission. In contrast, this prior connection is unnecessary in the second case. Starting from this, it can be said that our protocol will be non-connection oriented since, in our case, communication between our entities will be more efficient and will consume fewer resources, a fact that is important since the greater the battery consumption in our UAVs, the lower our flight autonomy.

The next step is to indicate which OSI model layers will be interesting for our designed protocol. At the physical layer level, VLC technology will be used since it allows for the sending and receiving of light pulses that will be detected as ones or zeros, depending on the on and off time of the transmitting LED. On the other hand, the data link layer will also be borrowed because using Ethernet allows for the formation of frames. On the other hand, a messaging protocol will be located at the application layer because it requires the capacity to manage and exchange messages. Thus, this proposed protocol will focus on the physical layer, VLC technology, and the application layer due to the capacity mentioned earlier.

As mentioned in the previous paragraph, the data link layer groups data bits into blocks called frames, which is perfect for VLC since it sends light pulses that can be interpreted as bits. This layer has three functions:

- The first one is the delimitation of frames through which the beginning and end of a data block can be known, thus allowing synchronization between sender and receiver.
- The second function it provides us is error control, which can ensure that the information received corresponds to the original issued.
- Finally, the third function is that it allows flow control so that it prevents the sender from saturating the temporary storage memory, or buffer, from receiving the destination due to the different speed or occupancy of the two parties.

In this way, it can be said that it has already been defined that the protocol to be developed will be a data link layer protocol; it must be noted that the "Machine-to-Machine" (M2M) protocol to be developed will be a bit-oriented protocol located in the data link layer, which will be layer two of the OSI model. Likewise, we chose a variable message format that allows us to send and receive messages of variable length, since not all messages are the same length.

Thus, it is already known that we will have frames, each being a block of information subdivided into fields where each will be used for a specific mission. Within our protocol, there will be a standard part for all the types of frames that will be present. Some fields that will be useful in each frame are defined in this common part. In Figure 12, you can see the common fields that will exist in all the frames of our protocol.

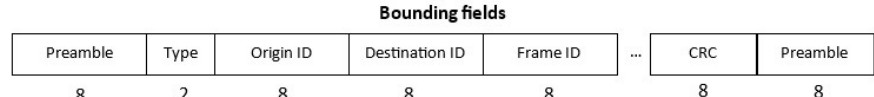

**Figure 12.** Format of the common parts of messages.

As we can see in Figure 12, a preamble allows us to know the beginning and end of a plot. This preamble will be made up of eight bits. Next, two bits appear that will indicate the type of message. With these two bits, it is indicated that we can have four types of messages:

- If the bits are 00, we are talking about a supervision frame in which we find relative information about which mode each detector UAV is in. These types of frames are exchanged between the detector UAV and the supervisor UAV, where the former informs the latter of what mode they are in, and the latter sends the information of the mode in which all the detector UAVs it supervises are in the base station.
- If bits are 01, these frames will notify the detector UAV at the traffic light of the need to make a mode change.
- In case the bits indicate 10, this signifies configuration frames. Their purpose is to facilitate adjustments to the thresholds for potential modes within the detector UAVs. This message originates from the base station and traverses through the supervisor UAV, with the latter being responsible for disseminating it to the relevant detector UAVs.
- Finally, for the value 11, these are acknowledgment (ACK) frames in which the sender is informed whether the message has been received and processed correctly or not. The decision to use an ACK protocol arises from the fact that we consider it important because the traffic lights have to work correctly, and an unwanted mode change does not occur if the information has not arrived correctly or has been affected. In case any unwanted mode change occurs, instead of managing traffic saturation to decrease it, we will help to do the opposite because more vehicles will arrive on the road and cannot get through it.

In Table 2, we can see a summary of the information presented in these last points, where we can see the types of frames that the protocol will support, the value that the bits that make up the frame type field will have, and the possible origins and destinations for each of the possible plots.

The next two fields we observe in Figure 12, both made up of eight bits each, that we find in the common part correspond to the source or origin ID and the destination ID. They will indicate the address of the sender of the frame as well as its receiver. As indicated above, there are eight bits available in each case, so the maximum number of elements that can make up our system will be 256 since, with eight bits, we can represent from the value 0 to 255. The maximum number will be the base station, the supervisory UAVs, the detector UAVs, and the traffic lights, so the sum of all these elements together should not be greater than that value.

**Table 2.** General information about frame types.

| Frame Type | Bits of Code | Origin–Destination |
|---|---|---|
| Supervision | 00 | UAV Detector–UAV Supervisor |
| | | UAV Supervisor– Base station |
| Mode change | 01 | UAV Detector–Traffic light |
| Configuration | 10 | Base station–UAV Supervisor |
| | | UAV Supervisor–UAV Detector |
| ACK/NACK | 11 | UAV Detector–UAV Supervisor |
| | | UAV Supervisor–Base station |

The following common field in the frame in Figure 12, again made up of eight bits, will be the frame ID, which will be used to know the frame number sent to affirmatively or negatively indicate to the sender that the frame has been received successfully. In this way, it can know if it is necessary to send the message again.

Following this frame ID field, the information will be sent and it will have a bit length that will vary depending on the type of message and the type of element that sends it. After the information fields, we can find the field that will be responsible for error detection and correction, the cyclic redundancy code (CRC). This CRC field, made up of eight bits, allows the receiver to know whether the integrity of the message has been received without any error. Finally, there is once again a preamble made up of eight bits to know the end of the plot. These fields that have been defined and have already been mentioned will be common to all the messages that will be exchanged between the elements of the system.

Next, we will explain what each of the four possible types of messages that will exist in the system will be like since we have seen that we have two bits to define the type of message, and this provides us with four types of messages.

- **Mode change frame**

  The first type of message that is exposed is the one that will be sent from a detector UAV to a traffic light, indicating that a mode change is required due to current traffic circumstances.

  In Figure 13, it can be seen that the two message-type bits will have the value "01". It is also observed that the first 34 bits form the common part in all frames. In the source ID field, we will have the address of the detector UAV that originated the message, while in the destination ID, the address of the traffic light will be entered. After these common fields, three bits appear that will indicate the operating mode in which the traffic light must operate. These three bits allow us to encode eight possible modes of operation, in our case having three possible free modes since, as we will see in Section 4.2, our system uses five modes of traffic lights and detector UAVs.

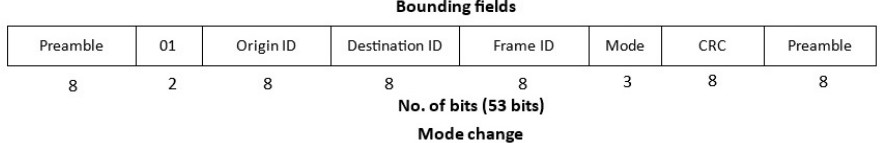

**Figure 13.** Mode change message format.

- **ACK and NACK frame**

  The next frame type that can be found would be the one in which we find that the two frame-type bits have the value "11". This type of frame, whose structure we can see in Figure 14, again has the 34 bits common to all the frames in our system.

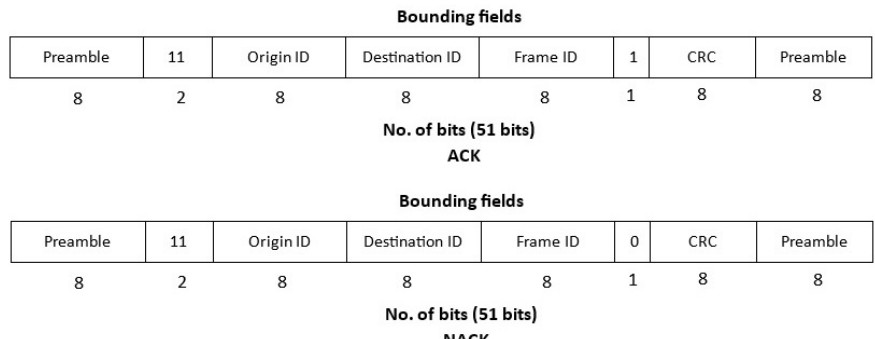

**Figure 14.** ACK/NACK message format.

It can be seen from Figure 14 that a single-bit frame field will define whether the ACK message has a positive result or a negative result. In this way, the sender can be informed that a frame has been processed successfully. This type of frame will be sent from the detector UAV to the supervisor UAV and from the latter to the base station and will confirm through the frame ID the type of message to which it is responding and whether it has been treated successfully or not.

- **Configuration frame**

  The third possible frame type will have the value "10" in the frame-type bits, corresponding to the configuration frame. As mentioned previously, these frames allow us to change the traffic congestion detection thresholds of the detector UAVs. These types of frames will be sent from the base station and are aimed at the detector UAVs with which we will have two types of configuration frames: those that will be sent between the base station and the supervisor UAV and the one that the supervisor UAV will send to the UAV-affected detector. As seen in Figure 15, the 34 common bits will again be present regardless of the sender of the frame.

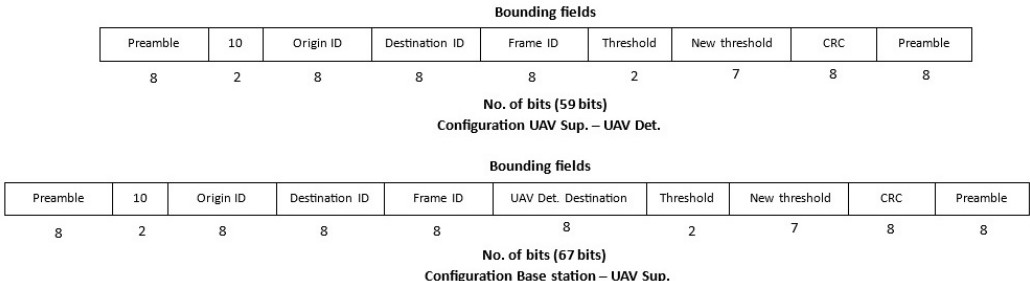

**Figure 15.** Configuration message format.

After that, depending on the sender and receiver of the frame, there are two possible frames. Knowing that the first part of this type of frame is sent from the base station to the supervising UAV, the lower type of frame in Figure 15 will be discussed first. The source ID will be this frame's base station, and the destination ID will correspond to the supervising UAV. As you can see, we will have eight bits that will indicate the address of the recipient detector UAV, two bits that will indicate which threshold must be modified in this detector UAV, and seven bits that allow us to indicate the value of this new threshold. As seen previously in Figure 10, there are different thresholds, which are $T_1$, $T_2$, $T_3$, and $T_4$, which correspond to the four possible values that the two threshold bits give us. That is, for "00" we will modify $T_1$, and with "01" it will be $T_2$, and so on. The new value to be inserted is given by the seven bits of the new threshold field, where we can represent values from zero to one hundred according to the scale in Figure 10 to be able to set the new value of the selected threshold.

Once the UAV receives the configuration frame from the base station, it must be sent to the affected detector UAV. To do this, we will now look at the upper frame of Figure 15. In this case, the source ID will be the supervisor UAV, and the destination ID will be

the detector UAV. After this, we see the two bits again to know which threshold must be modified and the seven bits that indicate the new value of the selected threshold.

- **Supervision frame**

  The last type of frame that we can find will be the supervision frame. In this case, two types of frames can be found, again, depending on the sender and the receiver. One type of these frames, which we can see in the upper frame of Figure 16, will be sent from time to time from the detector UAVs to the supervisor UAV. These mentioned frames will include in the source ID the detector UAV that sends it and in the destination ID the address of the supervising UAV. After this, there are three bits that allow the supervising UAV to be notified as to which mode the detector UAV that sent the message is in.

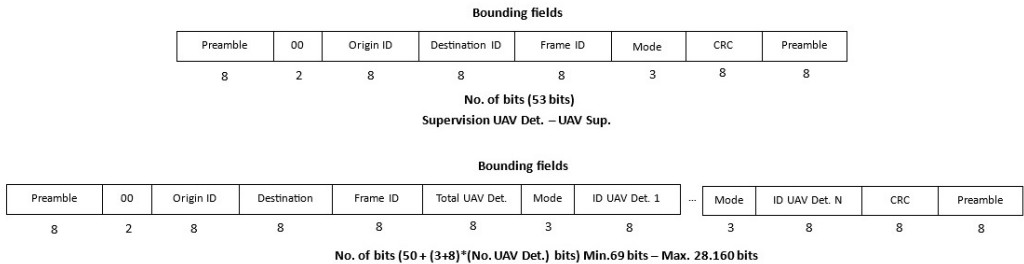

**Figure 16.** Supervision message format.

The other category of frames in this scenario pertains to the transmission from the supervisor UAV to the base station, providing information about the operational modes of all detector UAVs within the system. This can be observed in Figure 16. Following the standard section designating the source as the supervisor UAV and the destination as the base station, there is an eight-bit field that communicates to the base station the total count of UAV detectors in the system. Subsequently, the mode and address of the initial UAV in the system are dispatched, allowing the base station to receive mode and ID data as specified by the supervising UAV. Depending on the number of UAV detectors in the system, the frame may contain up to 28,160 bits. This does not present an issue during transmission, as it would take 1402 milliseconds to transmit at a rate of 2 Mbits per second.

Once the frame structure of our protocol is shown, let us examine some examples. In Figure 17, you can observe the designated directions for each component, indicated in green and blue. In this specific scenario, the detector UAV is relaying to the traffic signal that a transition to mode two of operation is advised due to the prevailing traffic conditions. The code bits, marked in red, signify a frame indicating a mode alteration. Furthermore, the frame number is highlighted in orange, while the three bits in brown provide instructions to the traffic light regarding the specific mode adjustment it should execute.

In the example depicted in Figure 18, the base station initiates the request, indicated by the address in brown, through the supervisor UAV (address in blue) for the configuration of threshold $T_2$. This request identifies the type of frame in red, while the threshold itself is indicated in gray, with a value of 30 represented in maroon. The request is directed to the detector UAV, which is identified by the green address. As in the previous example, the frame identifier is displayed in orange, which is employed for acknowledgments, as we will soon discuss.

However, in this case, due to a problem, the frame is either not received correctly or encounters a reception issue at the detector UAV. Consequently, a NACK (Negative Acknowledgment) message is returned, wherein we can observe the type of frame in red and the frame identifier in orange. This pertains to the field containing the value associated with a NACK frame, displayed in gray. The NACK frame is then relayed to the base station once more via the supervising UAV.

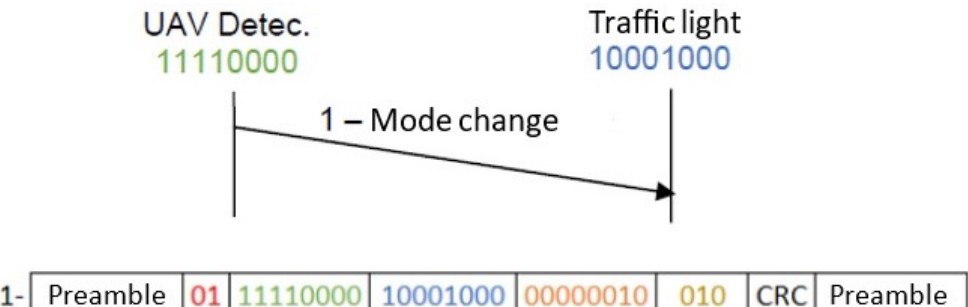

**Figure 17.** Example of mode change frame.

Upon re-issuing the request in the form of a configuration frame, the frame type and colors remain consistent, with the exception of the frame identifier, which is now different. This time, the alteration is successfully implemented, and a notification confirming the change is sent back to the base station. The frame successfully identified corresponds to this new frame that was dispatched.

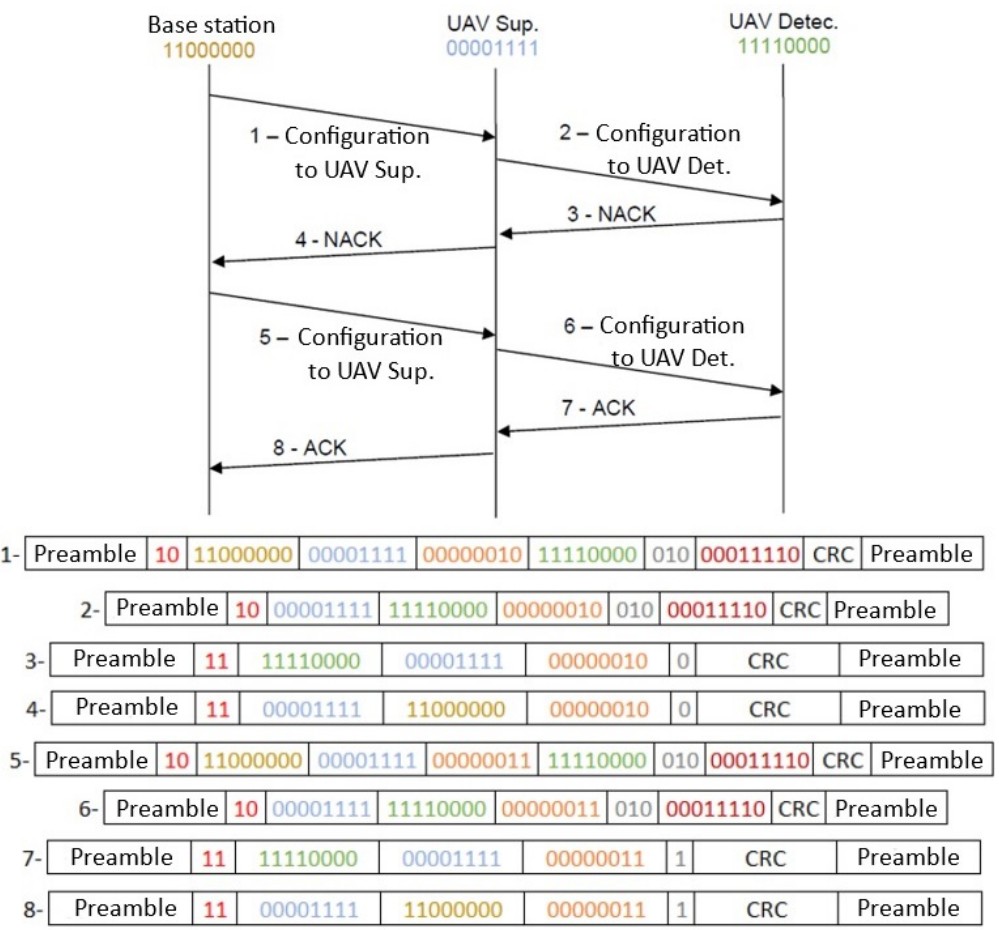

**Figure 18.** Example of configuration and ACK/NACK frames.

Finally, let us examine an example of supervision frames as illustrated in Figure 19. Here, we observe two detector UAVs whose addresses are denoted in green and yellow, respectively, conveying information about their respective states to the supervisor UAV, marked by the blue address, using supervision frames indicated in brown. Following this, the supervisor UAV dispatches a supervision frame to the base station, identified by the light brown address. The frame contains details about the number of detector UAVs in the system, as indicated in the field of the third light blue frame, along with their respective

states, as depicted in brown in the three plots. In this instance, one UAV is in mode two, while the other is in mode three.

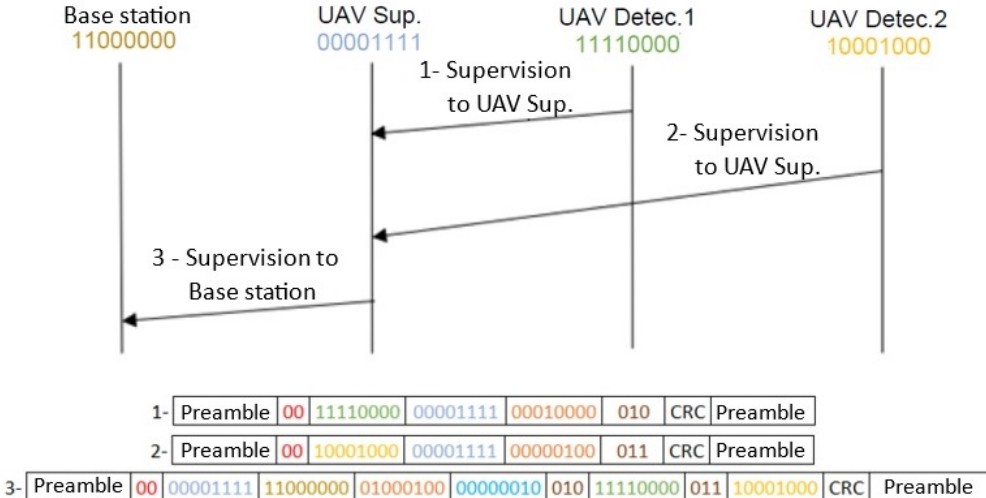

**Figure 19.** Example of supervision frame.

*4.2. Communication Protocol*

In this subsection, the ways in which we can find all the elements of our system will be introduced and explained. The modes of the supervisor UAV, detector UAV, and traffic light will be explained, going into a detailed explanation of what will be carried out and what conditions must be met for changes to be made. Regarding the dependency between operating modes that exist between the traffic light and the detector UAV, a scheme will be made to understand their interconnection and how the change in UAV mode can affect the traffic light.

- **Supervisor UAV:** This system element primarily operates in the "Supervision" mode and "Configuration" mode. In the former mode, information flows from the supervisor UAV to the base station, whereas in the latter, the supervisor at the base station receives the information and subsequently transmits it to the detector UAVs as required. The associated operating modes of the UAV supervisor can be observed in Figure 20.

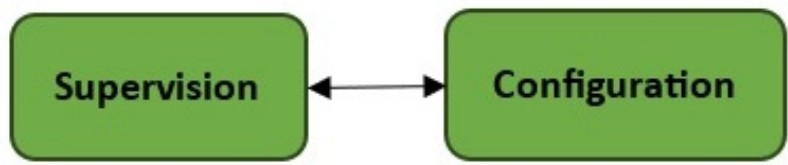

**Figure 20.** Supervisor UAV modes.

In the "Supervision" mode, the supervisor UAV transmits information regarding the status and mode of the detector UAVs under its supervision to the base station. This mode involves the reception of periodic information messages from the detector UAVs, and the data received from them are subsequently relayed to the base station. These messages, originating from various detector UAVs, will be intermittently received and promptly forwarded to the base station. As the supervisor UAV continually receives information from the detector UAVs, a consistent stream of data will be dispatched to the base station, ensuring real-time monitoring of the system's status.

- **Detector UAV:** The detector UAV has two main operating modes: "Radar" mode and "Normal" mode. Within the "Normal" mode, you can find four operating submodes, depending on the traffic detected, which are "Low", "Medium", "High", and "Extreme". Therefore, the detector UAV can be in "Radar" mode if the traffic crossing the road is minimal or non-existent and can change to "Normal-Low" mode as soon

as the traffic intensity exceeds the congestion threshold value for the change. When any mode change occurs in the detector UAV, it notifies the traffic light that it must also make a change to adapt. Likewise, if the detected traffic intensity is greater than the upper threshold value of the "Normal-Low" mode, it will switch to the "Normal-Medium" mode, and the traffic light will be informed that it must make a mode change. In Figure 21, we can see the schematic diagram of the modes in which the UAV detector can be found.

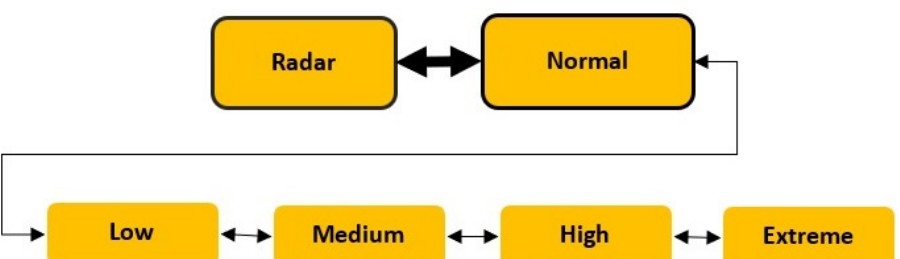

**Figure 21.** Detector UAV modes

Let us remember that the detector UAV is responsible for taking photographs in order to determine traffic congestion, so when the mode is changed the time that will pass between taking one image and the next will vary, time being higher for the "Normal-Low" mode and decreasing until reaching the minimum time that would correspond to the "Normal-Extreme" mode.

On the other hand, and as we have seen in the UAV supervisor section, the detector UAV sends supervision messages to the supervisor UAV from time to time with the objective of informing that the system is operating correctly. These messages are sent periodically to the UAV supervisor and always have the same time interval between them.

When the UAV is in "Radar" mode, it can be said that it is in a state similar to a low-consumption state. In this mode, the UAV does not take photographs since it is assumed that the traffic that travels along the road is minimal. The detection of the passage of vehicles is carried out using the technology provided by VLC. That is, through the light beam that is sent, we will detect the passage of possible vehicles. In this way, if the continued passage of vehicles is detected, it will go to "Normal" mode.

Regarding the "Normal" mode, as we have already mentioned, the UAV could be in four possible submodes, the first submode being the "Normal-Low" mode, which it reaches after leaving the "Radar" mode.

In the "Normal-Low" mode, the detector UAV will take photographs from time to time, with the time interval between images being the longest of all the submodes. When the UAV is in this mode, it is assumed that there is a light or low traffic load on the monitored road. As long as the image traffic congestion parameter value is contained between this mode's upper and lower threshold values, the detector UAV will not make any changes and will remain in "Normal-Low" mode, taking and processing the images. When the measured parameter is below the lower threshold of the mode, the UAV will go to "Radar" mode and will notify the traffic light that it must proceed to a mode change. On the contrary, if the measured congestion parameter's value exceeds the mode's upper threshold value, the UAV will go to the "Normal-Medium" mode and inform the traffic light that it must change to the specific associated mode.

When the UAV is in the "Normal-Medium" mode, the interval between photographs will be less than in the previous mode. In this way, the UAV can detect changes in traffic faster and more agilely and inform the traffic light so that it can adapt similarly. If the detector UAV is in this mode, the images will be treated in the same way so that depending on the intensity of the traffic detected on the supervised road and the traffic congestion parameter detected, the UAV may go to a "Normal-Low" mode

if the measured parameter is at a value lower than the threshold of the current mode, or to a "Normal-High" mode if, on the other hand, the measured parameter is higher than the upper threshold value of the mode. As with the other operating modes, if the value of the measured congestion parameter is between the upper and lower threshold values, the UAV will remain in the current mode.

Thirdly, we can examine the "Normal-High" mode. In this mode, the image-taking interval of the UAV will be reduced again, with the aim being that the system is capable of detecting changes in traffic and is able to adapt. As in the other modes, the image is treated identically, and the road congestion level will be extracted from it. Depending on this congestion parameter extracted from the image, the UAV can determine if it must change or stay in the same mode. If the parameter is below the threshold defined for this mode, it will switch to the "Normal-Medium" mode, while if the measured value of the parameter is greater than the upper value of the threshold, it will switch to the "Normal-Urgent" mode.

Lastly, there is the "Normal-Urgent" mode. This mode has the lowest time interval between taking photographs since it is assumed that there is very high congestion or saturation in traffic and that, if measures are not taken in time, this can lead to large delays. The detector UAV will be in this mode whenever the congestion parameter measured in the image exceeds the lower threshold value for this mode. Once the number of vehicles on the road is reduced and the image congestion parameter is below the threshold of this mode, the UAV will switch to the "Normal-High" mode and will inform the traffic light so that it also adapts its mode to the amount of traffic detected.

Finally, if the detector UAVs must be removed due to low battery, they inform the traffic light that it must make a mode change, since, in this case, the detector UAVs will stop monitoring the traffic and the control of the traffic light lights must be carried out by another device, which will be the traffic light regulator.

- **Traffic light:** The last element of the system that appears to us will be the traffic light. As indicated throughout this work, this element will be a passive element of the system since it will only execute the corresponding changes marked by the detector UAV.

  Like the detector UAV, the traffic light operates in two primary modes: "Autonomous" and "Normal" modes. Within the "Normal" mode there are four submodes, categorized based on the detected traffic intensity: "Low", "Medium", "High", and "Extreme". The organization of potential states for the traffic light is illustrated in Figure 22. The traffic light requires instruction from the detector UAV for a mode transition. Each submode within the "Normal" mode dictates varying durations for which traffic management lights are active. For instance, in the "Normal-Low" mode, the green light remains on for a shorter duration compared to the standard "Normal" mode. Conversely, in "Normal-Medium," the red light's duration is extended relative to the first mode. This adjustment in traffic light timings aims to facilitate efficient and agile traffic management.

  When the traffic light is in "Autonomous" mode, the order to turn the traffic light on and off will be regulated by the traffic light regulator to which it is connected. Thus, in "Autonomous" mode, the control of the traffic light will be, despite the redundancy, autonomous, just as it is currently being done in cities.

  Once the traffic light receives the message from the UAV indicating that it must go to "Normal" mode, the traffic light will disconnect the orders to turn the lights off and on from the traffic light regulator so that until it receives an order, the UAV will not make any further mode changes.

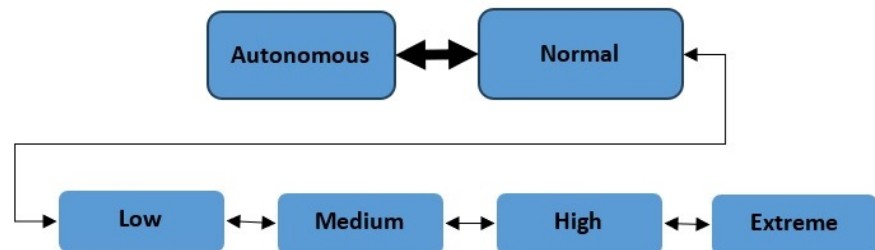

**Figure 22.** Traffic light modes.

When the traffic light receives the order to change to "Normal-Low" mode from the detector UAV, its lights will change the ignition time of each of them to adapt to the new situation. In this mode, the red light will have a shorter time while the green light will increase the time. This mode is associated with a low level of traffic with few vehicles on the road. The traffic light will remain in this mode until further orders from the detector UAV.

If the traffic light is in "Normal-Medium" mode, it will again make an adjustment to its ignition time. In this case, the red light time will be reduced again and the green light time will be increased. In this way, we will have a longer green light time than in the previous mode, and thus, traffic will be more fluid. In this mode, it is assumed that the level of vehicles on the road is not very high and that there is fluid traffic.

If the traffic light is in the "Normal-High" mode, it once again makes an adjustment to the lighting time of its lights. Again, the time that the red light is on will be reduced while the time that the green light will remain on will increase. It is assumed that if we are in this mode, there will begin to be traffic jams and crowds of vehicles on the road, so we should try to reduce the saturation on the road.

As the last possible mode of the traffic light, we find the "Normal-Extreme" mode. The traffic light will be in this mode when the UAV detects that there is a large congestion of vehicles. In this mode, a final adjustment is made to the time that the lights remain on, reducing the time of the red light to a minimum and increasing the green light to a maximum. In this way, we will be able to give a quick exit to vehicles that are on the saturated road.

As we have said in the text related to the detector UAV, if the supervising UAV notifies the detector UAVs of the withdrawal order, they inform the traffic light that it must make a mode change. In this case, the traffic light will return to "Autonomous" mode since UAVs provide no traffic congestion detection service, and the traffic light regulator will have to take care of the time that the lights are on for.

- **Relationship between the modes of the UAV detector and the traffic light:** Now that the modes in which the elements of the system can be found in are known, it is important to know what relationship exists between the detector UAV and the traffic light since, as we have been saying throughout this section, when the detector UAV considers a mode change, it must inform the traffic light of the change to the desired mode. It can be said that when the detector UAV changes modes, the traffic light must also do so, as indicated. Changing modes affects both differently because, as we have seen, in the case of the UAV it will influence the time interval between taking one image and the next, increasing as it goes, detecting greater congestion on the road. For its part, at the traffic light, as the detector UAV detects an increase in traffic, the time that its red light remains on will be reduced and the green light will remain on longer in order to allow vehicles to pass.

  In Figure 23, we can see on the left the possible modes of the traffic light and on the right all the possible modes of the detector UAV. Looking back to Figure 23, we can see the two main modes of operation of both elements and, separated by a dotted line, the submodes of each of them. Likewise, we can see that there are horizontal lines that join each of these modes and submodes. If submodes are joined by a horizontal line, this mean that they are associated. When the detector UAV is in that mode, it will

indicate to the traffic light that it must change to the associated mode. For example, if we imagine that the UAV is in "Normal-Low" mode, the traffic light will be in "Normal-Low" mode. The UAV detects the need to make a mode change and switches to "Normal-High" mode because there begins to be many vehicles on the road; it will switch to "Normal-Stop" mode and inform the traffic light that it should also switch to "Normal-Stop" mode. In the same way, if it is detected that traffic has decreased and conditions imply a change from "Normal-High" to "Normal-Low", the UAV will change to the latter mode and will once again inform the traffic light of the need for the change from "Normal-High" to "Normal-Low".

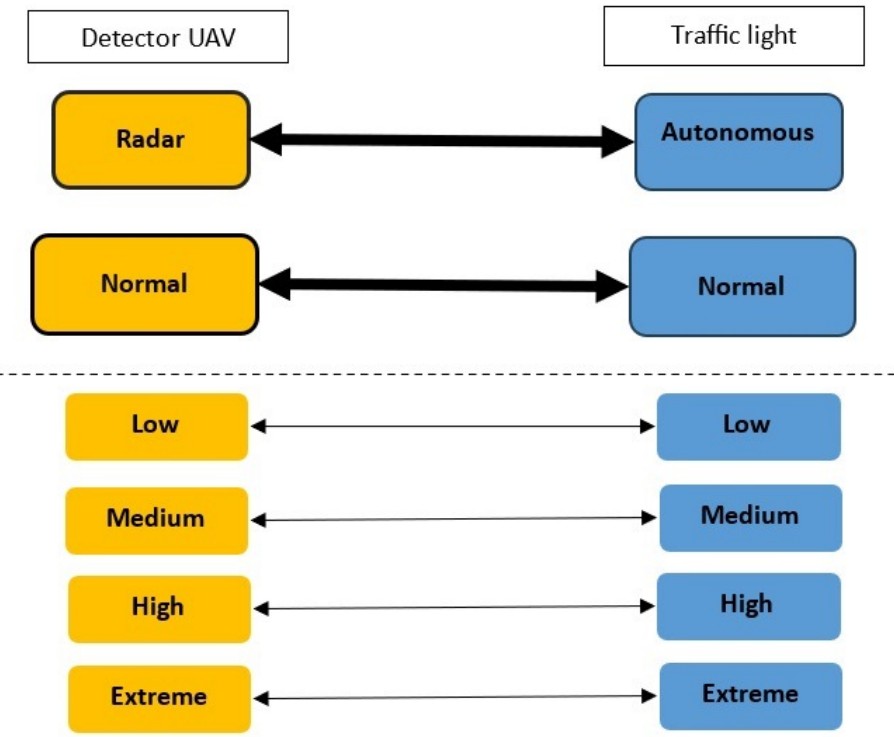

**Figure 23.** Relationship between detector UAV and traffic light modes.

So, we can affirm that, thanks to these associated modes between both elements, it will be possible to know, due to the supervision messages sent by the detector UAVs, the state of the traffic light since, by knowing the UAV's mode, we know the associated state of this second element that regulates traffic.

## 5. Prototype Insights Evaluation

Throughout the sections, we introduced a prototype system designed to advance traffic management with UAVs and contribute to developing smart cities with heightened sustainability and efficiency. Emphasizing a holistic approach, this prototype prioritizes system integration as its core focus in tackling urban traffic challenges. This integrated solution is further enriched by introducing a pioneering communication protocol and messaging system, collectively representing an innovative step forward in enhanced urban traffic management.

### 5.1. Uav and VLC Integration Analysis

The amalgamation of UAV and VLC components in our system introduces a paradigm shift in traffic management, bringing advantages that cater to the evolving needs of smart cities, yet it is not without its challenges warranting exploration. For example:

1.  Adaptability to Urban Environments: a VLC's non-directional nature, combined with UAV mobility, allows for flexibility in navigating complex urban landscapes, improving the system's adaptability to diverse city environments.
2.  Reduced Infrastructure Dependency: this integration reduces reliance on traditional ground-based infrastructure for traffic monitoring, potentially lowering costs and improving system scalability.
3.  Enhanced Communication Resilience: a VLC's use of the visible light spectrum provides an alternative communication framework that can be resilient to electromagnetic interference, offering a robust solution for communication in congested urban settings.
4.  Quick Deployment and Maneuverability: UAVs offer rapid deployment and maneuverability, allowing for swift adaptation to changing traffic patterns and emergencies, making the system highly responsive.
5.  Environmental Sustainability: these technologies support smart city initiatives, contributing to a more sustainable environment through improved traffic flow and reduced fuel consumption.

Moreover, exploring non-coherent communications, where channel estimation is deemed unnecessary, is suggested as a promising avenue for exploring novel modulation schemes. The lack of channel state information plays a crucial role in mitigating challenges like Doppler effects, ensuring a significant enhancement in communications, particularly in highly mobile environments, as exemplified in the UAV scenario discussed in this study. Instances of non-coherent schemes are illustrated in [25]. These schemes can even be expanded using satellites to increase coverage within the city without the need for terrestrial infrastructure that increases the cost of deployment [26].

*5.2. Messaging System Analysis*

The messaging system has been designed based on a philosophy of variable formats for your messages. This feature has or offers a series of benefits.

1.  Adaptability: messages are adapted to diverse content needs. They can accommodate short, concise messages and longer, detailed communications, providing flexibility for various contexts.
2.  Efficient Communication: variable-length messages enable efficient communication by tailoring the length of the information conveyed. This provides savings in bandwidth and resource savings that facilitate exchange.
3.  Enhanced Clarity: the flexibility in message length allows for clearer communication. Short messages are straightforward to grasp, while longer messages can provide in-depth explanations and details, enhancing overall clarity.
4.  Customization: variable-length messages permit customization based on the audience, platform, or communication channel. This adaptability ensures that messages align with the preferences and expectations of the intended recipients.
5.  Optimized Attention: short messages are often more attention-grabbing and suitable for quick consumption, effectively capturing immediate interest. Longer messages, conversely, cater to audiences seeking in-depth information, optimizing attention for different user preferences.
6.  Versatility: variable-length messages offer versatility in communication strategies. Whether aiming for brevity in certain situations or depth in others, this adaptability supports diverse messaging goals and communication styles.
7.  Balanced Information Delivery: longer messages allow for a more comprehensive topic exploration, ensuring the information is balanced and well-rounded. Variable length facilitates striking the right balance in delivering the intended message.

*5.3. Challenges and Future Trends*

Building upon the preceding analysis and the distinctive features inherent in this prototype, several lines of research emerge as both challenges and future trends. This aligns with the intention to introduce this design as a foundational stepping stone for

further exploration in research. The enumeration of challenges to be considered, including an exploration of other contemporary emerging techniques, forms a comprehensive outlook below:

1.  VLC Technology Challenges: VLC technology introduces a notable challenge, as VLC lacks the directional characteristics of a laser, relying on LED lights for information transmission. This raises potential interference from various urban light sources, such as streetlights and advertising panels. Future research will investigate the impact of light pollution on the communication protocol, determining the system's efficiency under different lighting conditions.

2.  Adverse Weather Conditions: a critical challenge arises in assessing the system's functionality under adverse weather conditions, such as rain or intense fog. The quality of the images captured by UAVs may be compromised, affecting vehicle detection accuracy. Future research will explore potential limitations and adaptations required for the system to operate effectively in varying weather scenarios.

3.  Image Processing Algorithm Optimization: the YOLO algorithm has been proposed for image processing in our prototype; however, alternative algorithms like SSD, Faster R-CNN, RetinaNet, or MobileNet-SSD present viable options. Future investigations will delve into a comparative analysis of these algorithms, aiming to optimize the on-board image processing capabilities of UAVs, thereby enhancing the efficiency of the traffic management system.

4.  Joint Sensing and Communication: a compelling avenue for future exploration involves the integration of joint sensing and communication, leveraging signals from existing or future communication systems for sensing purposes. This interdisciplinary approach opens up possibilities for our proposed system to seamlessly integrate with UAV onboard communication technologies, presenting an intriguing challenge and area for further research.

5.  UAV Components Challenges: Processing Power vs. Flight Durability for battery consumption supposes a commitment to refining and optimizing this delicate balance without isolating specific UAV components. This encapsulates a dedication to advancing the overall system's efficiency, emphasizing the integration and synergy of various elements rather than a detailed analysis of individual UAV components. This focus on the system's holistic functionality lays the groundwork for future research endeavors to enhance the integrated system's overall performance and endurance.

This prototype serves as a foundation document, outlining the challenges encountered and delineating future lines of research. The identified challenges provide a roadmap for refining the proposed system, ensuring its robustness and adaptability in diverse urban environments.

## 6. Conclusions

In this paper, we have introduced the prototype for a traffic management system that leverages a fusion of UAV and VLC technologies. This system demonstrates the ability to dynamically adjust its management strategies in response to varying road occupancy levels. The system's adaptability to road occupancy, facilitated by our meticulously designed messaging and communication protocols, holds promise for more efficient urban mobility solutions.

In terms of future research directions, a compelling avenue lies in incorporating additional sensors aboard the UAVs. These sensors could offer valuable data for smart cities, potentially revolutionizing urban planning and traffic management strategies. To accommodate this influx of new data, there may be a need to refine and expand our messaging protocol, ensuring it can effectively transmit the information collected by these supplementary sensors. This enhancement could significantly amplify the system's capabilities and contribute to even more sophisticated and responsive traffic management solutions in the smart cities of tomorrow.

**Author Contributions:** Conceptualization, V.M.B. and J.G.G.; methodology, V.M.B.; validation, V.M.B. and J.G.G.; formal analysis, V.M.B.; investigation, V.M.B. and J.G.G.; writing, V.M.B. and J.G.G. All authors contributed to the ideas that generated the original paper proposed herein; reviewing the manuscript, V.M.B. and J.G.G. All authors have read and agreed to the published version of the manuscript.

**Funding:** This research received no external funding.

**Data Availability Statement:** Data are contained within the article.

**Conflicts of Interest:** All authors declare that the research was conducted in the absence of any commercial or financial relationships that could be construed as a potential conflict of interest.

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
