# Peer review of "Enhancing Urban Mobility through Traffic Management with UAVs and VLC Technologies"

_drones, doi:10.3390/drones8010007_

Round 1

Reviewer 1 Report

Comments and Suggestions for Authors

Although the paper is well written with proper organization, I still feel that to step ahead, the authors must improve the following weaknesses.

This work presents enhancing urban mobility through traffic management with UAVs and VLC technologies. whereas:

               1.      The details of Visible Light Communication (VLC) are missing in the introduction section.

               2.      YOLO processes frames at the rate of 45 fps (larger network) to 150 fps (smaller network), which is better than real-time. In contrast, it has comparatively low recall and more localization error compared to Faster R_CNN. So, what motivates authors this selection in the presence of SSD, Faster R-CNN, RetinaNet, and MobileNet-SSD?

               3.      In 3.2. Traffic congestion or saturation detection: Figure 7 and Figure 9 are not relevant for Traffic congestion. It would be better if the drone-taken images explained parameters in the presence of congestion scenarios.

               4.      NACK-free protocol may assist you in less processing.

               5.      The authors did not discuss the battery constraint regarding processing and flight maintenance.

               6.      In figures, the font style is not the same.

               7.      How authors manage traffic using Unmanned Aerial Vehicles (UAVs) in foggy environments.

               8.      I could not find the result section. The inclusion of a results section holds significant importance in any research paper. Lacking a results section creates a sense of ambiguity for the reader, leaving them uncertain about the study's conclusions. Authors should recognize the pivotal role of the results section in contributing to the scientific discourse and take care to include this crucial element in their work.

               9.      This could be either a review article enabling authors to compare various existing models or a technical article allowing authors to present proposed results in various scenarios with tests and validation through a comparison with existing ones.

           10.      There are some grammatical mistakes i.e., interjections and conjunctions.

Comments on the Quality of English Language

Minor editing of the English language is required such as there are some grammatical mistakes, i.e., interjections and conjunctions.

Author Response

We sincerely thank the Reviewer for their valuable feedback and insightful suggestions. We truly appreciate your detailed review and are committed to addressing the concerns raised. Our responses to each point are in the attached document.

Reviewer 2 Report

Comments and Suggestions for Authors

A UAVs and VLC communication framework for Enhancing Urban Mobility was proposed in this manuscript. Some comments:

  1. The proposed frame work and methods were not verified, suggest to add evaluations or experiments and analysis for the results.
  2. The contributions other than UAVs and VLC communication framework can be summarized in the abstract part.
  3. The problems and challenges faced need to be discussed according to the research of this manuscript.
  4. The methods about how to manage the traffic lights within a big area can be discussed further. 
  5. The network types and connections illustrated in Figure 3 and Figure 4 can be merged into Figure 1. Also, Figure 5 and Figure 6 provided not so much information, but occupied space, reconsider how to represent the information of Figure 1-Figure 6 within one or two figures?
  6. The definition of the intensity and the calculation method need to be illustrated, and the occupancy or congestion calculation method is not clear, suggest to provide and reveal the contributions.
Comments on the Quality of English Language

  1. There is repeated works in Line 125, please check the typos and repeated sentences thoroughly. Minor editing of English language required.

Author Response

(The authors gave the same response as above.)

Reviewer 3 Report

Comments and Suggestions for Authors

In this paper, the authors present the challenges in future traffic and mobility, proposing a solution based on the UAV equipped with VLC technologies. The structure and presentation are well organized; however, there are a few points that need to be more clear.

1) It could be better to add more practical (real urban traffic) data-based simulation results, at least with some case studies.

2) The machine learning part of the paper is to some degree well written; however, there seems to be insufficient optimization methods in practical VLC-UAV operation.

3) The proposed approach assumes that fully networked VLC on the UAV is feasible. However, there are significant issues such as onboard processing, alignment problems, and SWap_C ... It could be beneficial to predict how those can be challenged so that subsequently the proposed method can be applied.

Comments on the Quality of English Language

In reference 11, DOI is wrongly written.

Author Response

(The authors gave the same response as above.)

Round 2

Reviewer 1 Report

Comments and Suggestions for Authors

Thank you for your comprehensive and accurate responses.

The clarity and logical coherence of the author’s explanations, coupled with their proactive approach to implementing essential revisions, have conclusively elevated the paper's overall quality. The feedback provided has been thoroughly addressed, particularly regarding:

1.      The emphasis placed on selection (YOLO)

2.      The improved clarity of figures

3.      The rationale behind the use of the ACK protocol

4.      The consideration of battery limitations

Comments on the Quality of English Language

Minor editing of English language still required.

Author Response

Thank you very much for your time in reviewing our responses and appreciating our work. Your feedback has been very constructive to improve the quality of the contribution.
We have carried out a new review to ensure any English grammatical errors. Also checked by a native colleague.

Reviewer 2 Report

Comments and Suggestions for Authors

No new comments for the revised manuscript.

Author Response

Thank you very much for your time in reviewing our responses and appreciating our work. Your feedback has been very constructive to improve the quality of the contribution.